# Reversing an Oncogenic Epithelial-to-Mesenchymal Transition Program in Breast Cancer Reveals Actionable Immune Suppressive Pathways

**DOI:** 10.3390/ph14111122

**Published:** 2021-11-02

**Authors:** Michelle M. Williams, Sabrina A. Hafeez, Jessica L. Christenson, Kathleen I. O’Neill, Nia G. Hammond, Jennifer K. Richer

**Affiliations:** Department of Pathology, University of Colorado Anschutz Medical Campus, Aurora, CO 80045, USA; michelle.m.williams@cuanschutz.edu (M.M.W.); sabrina.hafeez@cuanschutz.edu (S.A.H.); jessica.christenson@cuanschutz.edu (J.L.C.); kathleen.oneill@cuanschutz.edu (K.I.O.); nia.hammond@colorado.edu (N.G.H.)

**Keywords:** triple negative breast cancer, epithelial-to-mesenchymal transition, immune suppression, immunotherapy, metastasis

## Abstract

Approval of checkpoint inhibitors for treatment of metastatic triple negative breast cancer (mTNBC) has opened the door for the use of immunotherapies against this disease. However, not all patients with mTNBC respond to current immunotherapy approaches such as checkpoint inhibitors. Recent evidence demonstrates that TNBC metastases are more immune suppressed than primary tumors, suggesting that combination or additional immunotherapy strategies may be required to activate an anti-tumor immune attack at metastatic sites. To identify other immune suppressive mechanisms utilized by mTNBC, our group and others manipulated oncogenic epithelial-to-mesenchymal transition (EMT) programs in TNBC models to reveal differences between this breast cancer subtype and its more epithelial counterpart. This review will discuss how EMT modulation revealed several mechanisms, including tumor cell metabolism, cytokine milieu and secretion of additional immune modulators, by which mTNBC cells may suppress both the innate and adaptive anti-tumor immune responses. Many of these pathways/proteins are under preclinical or clinical investigation as therapeutic targets in mTNBC and other advanced cancers to enhance their response to chemotherapy and/or checkpoint inhibitors.

## 1. Introduction

Triple negative breast cancer (TNBC) is an aggressive breast cancer subtype that lacks expression of targetable receptors maintained by other subtypes of the disease, including estrogen receptor (ER), progesterone receptor (PR) and human epidermal growth factor receptor-2 (HER2). As a result, few targeted therapies are approved for use in a majority of TNBC cases. Most patients are treated with a combination of chemotherapy, radiation and surgery. Further, TNBC recurs as metastatic disease more rapidly than the other breast cancer subtypes, usually within 2–3 years after diagnosis [1]. The 5-year survival rate for patients with metastatic TNBC (mTNBC) is only 26% [2]. Approval of the first immunotherapies in mTNBC, the checkpoint inhibitors pembrolizumab (blocking antibody against programmed death-ligand 1/PD-L1) [3] and atezolizumab (blocking antibody against programmed cell death protein-1/PD-1) [4], generated enthusiasm for the use of immunotherapies in breast cancer. However, the 2-year overall survival for mTNBC patients treated with the combination of chemotherapy and checkpoint inhibitor was only ~50% for pembrolizumab [3], and atezolizumab was voluntarily withdrawn from use in mTNBC for reasons unrelated to safety or efficacy. These clinical findings suggest that immunotherapy for mTNBC could be improved by combining checkpoint inhibitors or identifying additional immunotherapy strategies. 

There is increasing justification for alternative immunotherapy approaches for mTNBC. Recent reports indicate that TNBC metastases are more immune suppressed than primary tumors because they have decreased abundance of many immune cell types including CD4^+^ and CD8^+^ T cells [5,6]. However, the one immune cell type that increased in mTNBC was suppressive macrophages [5,6]. These studies suggest that the stress of metastasis may alter TNBC and its interaction with the tumor microenvironment (TME) leading to enhanced immune suppression. Thus, an improved understanding of immune modulatory pathways increased during metastasis may reveal alternative immunotherapy approaches for TNBC, including therapies that limit suppressive macrophages. Our group and others have approached this idea by manipulating oncogenic epithelial-to-mesenchymal transition (EMT) programs in TNBC models. This revealed pathways utilized by TNBC to support immune suppression and metastasis. Each section of this review will focus on one pathway identified by these studies, including those involved in tumor cell metabolism, cytokine milieu and secretion of additional immune modulators. Discussion will focus on current evidence from preclinical and clinical studies that suggest targeting these pathways could limit TNBC metastasis through impacts on both TNBC cells and cells in the TME.

## 2. Experimental Models of Breast Cancer Metastasis and Epithelial-to-Mesenchymal Transition (EMT)

### 2.1. Modeling Breast Cancer Metastasis

#### 2.1.1. Immune-Competent Models of Breast Cancer Metastasis

The strengths and weaknesses of breast cancer metastasis models have been thoroughly reviewed by other groups [7,8,9]. This section will briefly highlight the advantages and disadvantages of the most common in vivo, immune-competent mTNBC models discussed throughout this review. A commonly used spontaneous lung metastasis model is the transgenic Mouse Mammary Tumor Virus-Polyoma Virus Middle T Antigen (MMTV-PyMT) model that recapitulates many aspects of human breast cancer biology [10]. Mouse mammary tumor cell lines are also frequently used. These are generated from spontaneously arising tumors (examples: 4T1, 66Cl-4 and 67NR cells developed from a tumor in BALB/c mice [11]) or transgenic models (example: Met-1 cells developed from a late-stage MMTV-PyMT tumor [12]). These cell lines have varying metastatic capacities and can be introduced orthotopically into the primary site (mammary fat pad) and progress to lung metastasis. In some cases, clones of mammary carcinoma cell lines that spread to other metastatic sites were generated by selection, including the 4T1.2 bone-trophic derivative of 4T1 cells [13]. Alternatively, mammary carcinoma cells can be injected into the tail veins of mice to preferentially seed the lung. However, the tail vein injection model does not include the early stages of metastasis, such as detachment from the basement membrane, invasion and intravasation. Further, this technique does not model the full immune response to metastasis, including formation of the pre-metastatic niche that relies on immune cell involvement. The advantage of tail vein injection is that its independent of primary tumor growth and thus can be a less variable and more reproducible model of late-stage metastasis [14]. To model metastasis to organs other than the lung, mammary carcinoma cell lines can be injected into other sites. For example, portal vein injection can be used to model breast cancer liver metastasis [15], and cardiac injection can be used to model more widespread metastasis, especially disease that spreads to the bone (reviewed in [16]).

#### 2.1.2. Patient Derived Xenograft Models of Breast Cancer Metastasis

The metastasis models discussed above use immune-competent mice which allows analysis of the immune system; however, many of these models only metastasize to select sites, primarily the lungs. To better model the heterogeneity of breast cancer metastatic spread, many groups have developed patient derived xenograft (PDX) models of breast cancer. PDXs are patient specimens that were introduced into mammary fat pads and passaged in immunocompromised mice. PDXs are often derived from metastatic sites such as pleural effusions and they better recapitulate the metastatic profile of donor breast cancer patients [17,18]. A disadvantage of PDX models is that immunocompromised mice lack many components of the immune system. As the number of clinical trials that include immunotherapies increase in breast cancer, more extensive testing of PDX models in mice reconstituted with human immune cells, termed “humanized” mice, is needed. This will determine whether these models can recapitulate or predict patient responses to single or combination immunotherapy treatments. 

### 2.2. Manipulating EMT in Breast Cancer Models

#### 2.2.1. Introduction to EMT

EMT is a normal developmental process during which epithelial cells detach from the basement membrane, lose their cell-to-cell junctions and transition to a more mesenchymal phenotype characterized by increased motility and invasiveness (as reviewed in [19]). TNBC cells aberrantly co-opt this process and undergo at least partial EMT to support pro-tumor phenotypes such as chemotherapy resistance, metastasis and immune suppression. A reciprocal mesenchymal-to-epithelial transition often occurs at the metastatic site to allow outgrowth. Thus, carcinoma cells exist in a continuum between epithelial and mesenchymal states, leading to the term “epithelial-to-mesenchymal plasticity” [20]. Both normal and oncogenic EMT are modulated by transcription factors including the Zinc finger E-box-binding homeobox 1 (ZEB1), SNAI1 (SNAIL), SNAI2 (SLUG) and Twist-related protein 1 (TWIST). Each of these function as transcriptional repressors of the well-known epithelial genes, including *CDH1* that encodes E-cadherin [21]. 

#### 2.2.2. Methods to Manipulate EMT in Breast Cancer Models

To study breast cancer EMT and its impact on immune suppression, some groups such as the Weinberg group overexpressed EMT transcription factors [22,23]. Additionally, they selected for a population of cells with high EMT-transcription factor expression. An alternative strategy promotes EMT in epithelial-like breast cancer cells, such as those derived from ER+ disease. In this case, well-established EMT inducers like transforming growth factor-beta (TGF-β) are used [24]. Interestingly, TGF-β impacts EMT partly through regulation of micro-RNAs (miRNAs) which are powerful post-transcriptional regulators of gene expression. Our group demonstrated that more mesenchymal-like human TNBC maintain a distinct miRNA profile compared to more epithelial-like ER+ disease [25], and the miR-200 family was the most differentially expressed. The miR-200 family members are known as the “guardians of the epithelial phenotype” because they target several mesenchymal transcription factor transcripts for degradation or translation inhibition. One family member, miR-200c, has been restored to mesenchymal-like TNBC models to effectively reverse EMT. In a claudin-low breast cancer model, miR-200c restoration decreased primary tumor growth and late-stage lung metastasis [26]. We also used miR-200c restoration to reveal mechanisms that support TNBC chemotherapy resistance and metastasis [27,28]. These studies demonstrate that manipulation of EMT through several mechanisms can be a powerful tool to identify pathways that support TNBC metastasis. 

## 3. EMT and Immune Modulation

### 3.1. Preclinical and Clinical Evidence for Immune Modulation via EMT in Breast Cancer

Clinical analysis of gene profiling conducted on 11 different cancer types, including the breast, led to generation of a pan-EMT signature that linked EMT to immune suppression [29]. Tumors with the highest mesenchymal-like scores had decreased expression of miR-200 family members and increased expression of genes encoding checkpoint proteins associated with suppressed T cells, such as *PDCD1* (PD-1) and cytotoxic T-lymphocyte associated protein 4 (*CTLA4*/CTLA-4). Ligands for checkpoint proteins were also correlated with a mesenchymal-like score, including *CD274* (PD-L1) that is expressed by tumor cells and several immune cells in the TME. However, tumors that retained a more epithelial-like profile expressed a gene signature associated with higher levels of the miR-200 family. We modeled these clinical findings by restoring miR-200c to human TNBC cells lines and the Met-1 mouse mammary carcinoma cell line. Restoration of miR-200c decreased expression of genes in the pan-EMT signature and those associated with immune modulatory pathways including allograft rejection, complement and cytokine signaling [30,31]. We further explored the relationship between miR-200c and immune modulatory pathways in breast cancer specimens from The Cancer Genome Atlas (TCGA) [32]. In this clinical dataset, miR-200c expression inversely correlates with genes representing similar immune modulatory pathways to those altered by miR-200c in our TNBC models, including cytokine signaling, allograft rejection and complement (Table 1). These findings suggest that high expression of miR-200c may predict a more active TME. To test this hypothesis, we conducted CIBERSORT analysis on the same TCGA dataset [32]. CIBERSORT is an algorithm that uses 22 established immune cell gene profiles to predict relative immune cell abundance from bulk mRNA sequencing data [33]. Breast cancer patients with miR-200c expression in the top quartile have a significant increase in T follicular helper (T_fh_) cells when compared to those with miR-200c in the bottom quartile (Figure 1A). T_fh_ are a marker of tertiary lymphoid structures (TLS), and TLS predict a better overall survival in breast cancer [34]. B cells are also an essential component of TLS; however, CIBERSORT predicted no change in the abundance of B cell populations (memory B cells and plasma cells) with miR-200c expression (Figure 1B). This finding demonstrates possible limitations of using bulk mRNA sequencing data to fully capture the complexities of the TME. Finally, CIBERSORT predicted a trend towards increased M1 anti-tumor macrophages in specimens with high miR-200c (Figure 1C), suggesting that the TME of miR-200c-expressing breast cancers may be more anti-tumor.

### 3.2. Manipulation of EMT in Breast Cancer Models Reveals Additional Immune Modulators

Given the link between EMT and immune suppression in breast cancer specimens, our group and others have manipulated EMT in TNBC models to identify additional clinically relevant immune modulatory pathways. We restored miR-200c to human TNBC cell lines and this decreased expression of known immune modulatory miR-200c targets [30] such as PD-L1 [35,36]. This study also revealed new immune suppressive miR-200c targets that may dampen anti-tumor immunity in mTNBC, including tryptophan-2,3 dioxygenase (TDO2), chitinase-3 like-1 (CHI3L1) and heme oygenase-1 (HO-1). Restoration of miR-200c to Met-1 mammary tumors derived from the MMTV-PyMT model enhanced secretion of immune modulatory cytokines, like granulocyte macrophage-colony stimulating factor (GM-CSF) [31]. In a study conducted by a separate group, miR-200c was restored to eight mammary carcinoma models [37]. This altered monocyte and neutrophil infiltration, possibly due to changes in secretion of cytokines such as macrophage colony-stimulatory factor (M-CSF). Seminal work by the Weinberg group utilized cell lines created from MMTV-PyMT tumors that were characterized as more epithelial versus mesenchymal-like due to expression of EMT transcription factors and epithelial markers such as E-cadherin [22]. When introduced into mammary fat pads, the mesenchymal-like cell lines generated more immune suppressed tumors when compared to epithelial-like counterparts [22]. Exploration of secreted factors that differed between these models revealed 5’-Nucleotidase (NT5E or CD73), M-CSF and osteopontin (OPN) were increased in mesenchymal-like tumors [23]. Knocking down each one of these factors in tumor cells enhanced response to anti-CTLA-4 antibodies. The remainder of this review will focus on the factors identified by EMT manipulation (CD73, HO-1, TDO2, GM-CSF, M-CSF, CHI3L1 and OPN) and will highlight preclinical and clinical studies that demonstrate how each impact breast cancer metastasis and immune suppression (summarized in Figure 2).

## 4. Modulation of EMT Reveals Immune Suppressive Enzymes

EMT metabolically rewires breast epithelial cells [38,39] and may contribute to the dynamic metabolic remodeling required during metastasis (as reviewed in [40]). Metastatic tumor cells that alter their metabolism may also extrinsically impact the TME (as reviewed in [41]). This section will focus on three enzymes, CD73, HO-1 and TDO2, that were identified by manipulating breast cancer EMT by our group [30] and the Weinberg group [23] and summarize how each may impact breast cancer metastasis and anti-tumor immunity.

### 4.1. 5′-Nucleotidase (NT5E)/CD73 

#### 4.1.1. CD73 and Breast Cancer Metastasis

Solid tumors often have high adenosine levels in their extracellular fluid [42]. This correlates with elevated expression of the rate-limiting adenosine producing enzyme NT5E, more commonly known as CD73, on multiple cells in the TME including tumor and immune cells. Adenosine production by CD73 requires the degradation of adenosine triphosphate (ATP) to adenosine monophosphate (AMP) by ectonucleoside triphosphate diphosphohydrolase-1 (CD39) (Figure 3). AMP is then converted to adenosine by CD73. Elevated adenosine and CD73 in breast cancer specimens correlated with poor prognosis, metastasis and resistance to chemo- and immunotherapy [43,44,45,46]. CD73 in breast cancer cells promoted EMT, migration and proliferation due to interactions with oncogenic signaling through the TGF-β, β-Catenin and mitogen-activated protein kinase (MAPK) pathways [23,47,48]. 

#### 4.1.2. Adenosine and the Breast Cancer Microenvironment

Adenosine binds to four adenosine receptors (A1R, A2AR, A2BR, A3R) on breast cancer cells to activate pro-tumor phenotypes like ER signaling, stemness and motility [49,50,51]. In one study, tumor cell-A2BR enhanced spontaneous 4T1.2 lung metastasis [52]. In a separate study using the same model, tumor-derived adenosine simultaneously supported tumor cell motility and metastasis while also decreasing anti-tumor natural killer (NK) cell activity [53]. Thus, tumor cell-adenosine may function in both an autocrine and paracrine manner to support tumor progression. In fact, the tumor-promoting effects of CD73 are believed to be primarily mediated by suppression of anti-tumor immune cells, such as T cells and NK cells. Adenosine can alter immune cell signaling through essential functional pathways such as interleukin-6 (IL-6), interferon-gamma (IFNγ) and arginase-1 (ARG1) [45,54,55,56,57,58]. Indeed, the Weinberg group showed that silencing tumor cell-CD73 effectively enhanced CD8^+^ T cell cytotoxicity [23], while other groups demonstrated that it dampened the activity of pro-tumor immune cells [46,57,58]. Treatment of TNBC-like 4T1.2 primary tumors or establish lung metastases with anti-CD73 antibodies enhanced sensitivity to doxorubicin and checkpoint inhibitors by activating the adaptive immune response [59,60]. These preclinical studies suggest that adenosine orchestrates a complex pro-tumor signaling network. Targeting this network may dynamically reshape the TME to be more anti-tumor. However, the impact of immune cell-derived adenosine on breast cancer cells needs to be further explored. For instance, a major source of tumor adenosine may be regulatory T cells (Tregs) that express elevated CD39 and CD73 upon activation (reviewed in [61]). The impact of Treg-produced adenosine on cancer cells or other cells in the TME has yet to be determined in TNBC. 

#### 4.1.3. Clinical Targeting of CD73 in Breast Cancer 

Given the preclinical findings that CD73 and downstream adenosine signaling have potent tumor cell intrinsic and extrinsic effects, this pathway is being explored clinically. One group generated an adenosine signaling gene signature that predicted decreased survival and poor response to immunotherapies in patients representing all cancers in the TCGA [62]. Thus, CD73 may be a biomarker for response to chemo- and immunotherapy [45,63,64,65]. Early clinical targeting of the adenosine pathway is underway in breast cancer, and preliminary results suggest treatment strategies are on-target and well-tolerated (reviewed in Table 2). However, emerging preclinical studies call for simultaneous inhibition of multiple parts of this pathway, such as inhibiting both CD39 and CD73 (reviewed in [63]). This approach needs to be explored further preclinically and clinically in breast cancer. 

### 4.2. Heme Oxgenase-1 (HO-1)

#### 4.2.1. HO-1 and Breast Cancer Metastasis

HO-1, the inducible heme oxygenase isoenzyme, degrades heme into carbon monoxide (CO), ferrous iron, and biliverdin that is quickly converted to bilirubin by biliverdin reductase (BLVR) (Figure 4). Through its catabolites, HO-1 plays an established role in responding to oxidative stress and maintaining cellular homeostasis. HO-1 is also implicated in tumor therapy resistance and metastasis. In breast cancer models, HO-1 was elevated in cells that survived or were resistant to chemotherapy treatment [69,70,71,72]. Competitive inhibition of HO-1 suppressed breast cancer cell growth [73] and decreased tumor growth in MMTV-PyMT tumors when used in combination with chemotherapy [74]. In clinical studies, HO-1 protein or mRNA (*HMOX1*) correlated positively with breast cancer progression [75,76]. *HMOX1* also predicted poor overall survival and decreased relapse free survival for breast cancer patients [74,77], demonstrating a role for HO-1 in tumor progression. However, HO-1 has a controversial role in EMT. HO-1 was elevated in mesenchymal-like TNBC versus epithelial-like ER+ breast cancer models [30], in part due to its regulation by miR-200c [78]. Other studies indicate that HO-1 inhibits breast cancer EMT [79,80], suggesting the reciprocal effects of HO-1 and EMT need to be further delineated.

#### 4.2.2. HO-1 and Immune Suppression in Breast Cancer 

HO-1 is emerging as an important modulator of anti-tumor immunity. Enzymatic HO-1 inhibition re-sensitized a resistant mammary carcinoma model of TNBC to anti-PD-1 treatment in obese mice [81]. Further investigation demonstrated that HO-1 protects tumor cells from T cell killing induced by PD-1 blocking antibodies [81]; however, the impact of this combination in lean mice remains to be explored. In the TME, HO-1 is also highly expressed in tumor-infiltrating myeloid cells [82]. Pharmacologic inhibition of HO-1 in 4T1 mammary tumors that are a model of TNBC reverted pro-tumor M2 macrophages to an anti-tumor M1 phenotype [73]. In a separate study, HO-1-expressing tumor associated macrophages (TAMs) supported mammary carcinoma lung metastasis by enhancing metastatic cell intravasation [83]. These studies suggest that HO-1 inhibition in breast tumors may simultaneously target tumor cells and immune cells to limit metastasis. 

HO-1 may additionally impact the TME via its enzymatic byproducts (reviewed in [84]). In models of organ transplant and autoimmune disease, biliverdin limited T cell, neutrophil and macrophage infiltration and proliferation [85,86]. Bilirubin impacted the function and proliferation of endothelial cells, macrophages, T cells and dendritic cells [87,88,89,90,91]. The HO-1 metabolite CO protected endothelial cells from apoptosis by upregulating pro-survival MAPK signaling [92]. These findings suggest that the byproducts of either tumor cell or macrophage HO-1 may impact the TME, calling for further investigation of this pathway and its inhibition in mTNBC. It is exciting to note that there are many clinical strategies to target HO-1. Some of the HO-1 enzymatic inhibitors used in the studies discussed here are FDA approved for use in newborns with severe jaundice, a disease characterized by high serum bilirubin levels (reviewed in [93,94]). Thus, HO-1 inhibitors could be repurposed as a treatment for mTNBC, although these clinical studies have not yet begun. 

### 4.3. Tryptophan 2,3-Dioxygenase (TDO2)

#### 4.3.1. TDO2 and Breast Cancer Metastasis

The first step of the multistep tryptophan (Trp) catabolism pathway, which ultimately leads to de novo synthesis of nicotinamide adenine dinucleotide (NAD^+^), involves the conversion of Trp to kynurenine (Kyn) by TDO2 or Indoleamine 2,3-dioxygenase (IDO) [95]. TDO2 is normally expressed in the liver and brain and is responsible for modulating circulating Trp levels that have numerous systemic effects [96]. IDO is expressed in most normal tissues and for this reason it has been more extensively studied in cancers (reviewed in [97]). However, inhibition of IDO did not improve response of melanoma patients to checkpoint inhibitors, possibly due to compensation by TDO2 (reviewed in [98]). These clinical results led to increased exploration of TDO2 in cancers, including in TNBC. Gene expression of TDO2 was higher in breast cancer when compared to normal breast tissue, and this upregulation was often associated with more mesenchymal-like, aggressive breast cancer subtypes [99,100]. Breast cancer patients with high TDO2 also had poor distant metastasis-free survival and overall survival when compared to those with low TDO2 levels [99,101,102]. These clinical findings suggest that EMT may enhance TDO2 to support breast cancer metastasis. Indeed, restoration of miR-200c to human TNBC cell lines directly targeted and repressed TDO2 [30]. Additional preclinical studies demonstrated that TDO2-produced Kyn enhances tumor cell proliferation, migration and invasion in an autocrine manner [99,101]. For example, we showed that TNBC cells grown in forced suspension cultures to model the anchorage-independent stages of metastasis had increased TDO2 expression and Kyn production [99]. Under these conditions, Kyn bound to tumor cell aryl hydrocarbon receptor (AhR) and supported cancer cell survival. Thus, TDO2 inhibition decreased the outgrowth of TNBC cells in the lungs of immunocompromised mice following tail vein injection. These studies demonstrate a role for TDO2-produced Kyn in TNBC progression; however, further analysis of the impact of metabolites downstream of Kyn on breast cancer cells is needed.

#### 4.3.2. Tryptophan Catabolism and the Breast Cancer Microenvironment 

The Trp catabolism pathway also has paracrine effects on the TME due both to the local depletion of Trp and accumulation of Kyn (reviewed in [102]). IDO-dependent Trp catabolism induced CD4^+^ helper T cell apoptosis [103,104], CD4^+^ T cell differentiation into suppressive Tregs [105,106,107] and CD8^+^ cytotoxic T cell dysfunction [108]. Together, these effects on T cells may suppress the adaptive anti-tumor immune response to support tumor progression; however, the impact of TDO2 on tumor immune evasion is underexplored in breast cancer. Initial studies from our group have begun to address this gap in knowledge by demonstrating that TDO2-dependent Kyn secretion reduced the viability and function of primary human CD8^+^ T cells [109]. Future studies should determine whether TDO2 can support an immune suppressed microenvironment by impacting multiple T cell types, as previously observed with IDO, through either depletion of Trp or production of Kyn and other downstream Trp metabolites.

#### 4.3.3. Clinical Targeting of Tryptophan Catabolism

Due to the tumor cell intrinsic and extrinsic roles of Trp catabolism, IDO inhibitors are being investigated in multiple clinical trials. Inhibitors of both IDO and TDO2 are also in development. Studies showed that TDO2 predicts a worse outcome in breast cancer patients than IDO [99,109]. This suggests that TDO2 may be more important than IDO in promoting breast tumor progression and calls for TDO2 inhibitors to enter clinical testing in mTNBC. However, TDO2 inhibitors are only under preclinical development (reviewed in [110]). The first TDO2 inhibitor ever developed, 680C921 [111], has been widely tested in cell culture models, but its limited bioavailability hindered in vivo studies [112]. A newer inhibitor, LM10, with improved solubility and bioavailability was developed for in vivo use and is currently under investigation [113]. Further, dual IDO and TDO2 inhibitors are being explored that should completely block tumor production of Kyn [110,114,115,116,117]. With a similar objective, Kynases, which are derivatives of bacterial enzymes, are being investigated for their ability to degrade Kyn in the TME [118,119]. While each of these treatments is promising, several issues need to be considered before clinical testing. For instance, immune evasion is not only caused by Kyn accumulation, but also by Trp depletion which will not be affected by Kynases. After more extensive preclinical testing, these treatments could provide an additional immunotherapy to limit mTNBC progression.

## 5. EMT Regulated Immune Modulatory Cytokines

Cancer cells secrete several cytokines to impact immune cell infiltration, differentiation, activation and function. Each group that manipulated EMT in TNBC models using miR-200c or EMT transcription factor levels identified changes in tumor cell cytokine secretion [23,31,37]. Interestingly, these reports revealed that EMT suppresses or activates a different cytokine milieu, which is possibly a result of heterogeneity in the breast cancer and EMT models used. This section will discuss some of the common cytokines identified, GM-CSF and M-CSF, and how they impact immune cells in the breast cancer TME.

### 5.1. Granulocyte-Macrophage Colony-Stimulatory Factor (GM-CSF)

#### 5.1.1. GM-CSF and Breast Cancer Metastasis

Restoration of miR-200c altered several cytokines in human breast cancer and mammary carcinoma models including GM-CSF [31], a potent inducer of monocyte differentiation, maturation and function. Other groups have similarly demonstrated that EMT reversal in TNBC-like mouse models via depletion of the EMT transcription factor *Snail* increased GM-CSF [120]. In this study, GM-CSF supported M1 anti-tumor macrophage polarization and treatment with a GM-CSF blocking antibody reversed macrophages to a pro-tumor M2 polarization state. This suggests that downregulation of GM-CSF may be a prominent mechanism utilized by mesenchymal-like breast cancers to modulate macrophages. In support of this idea, intra-tumoral delivery of GM-CSF to orthotopic Met-1 mammary tumors decreased primary tumor growth, lung metastasis and M2 macrophage polarization by enhancing M1 macrophage polarization [121]. However, a recent study showed that mammary tumor cell produced GM-CSF supports the presence of suppressive myeloid cells [122]. This could be because GM-CSF induces myeloid derived-suppressor cells (MDSCs) (reviewed in [123]), a pro-tumor immune cell type that dynamically suppresses T cell responses [124]. Since MDSCs express many of the same markers as both M1 and M2 macrophages [125], further characterization of the impact of GM-CSF on MDSC populations during TNBC EMT is needed. Other studies also suggest that mesenchymal-like breast cancer cells preferentially secrete GM-CSF [126,127]. These findings demonstrate that each breast tumor may secrete a different cytokine milieu. Further investigation of cytokine profiles between breast cancer patients and metastatic sites is needed.

#### 5.1.2. Clinical Testing of GM-CSF Therapy 

In addition to impacting macrophages, GM-CSF activates anti-tumor T cell responses by enhancing dendritic cell (DC) antigen presentation. Initial studies in the late 1990s demonstrated that a tumor vaccine generated from irradiated GM-CSF-overexpressing mammary carcinoma cells provided 100% protection against mammary tumor formation [128]. Later studies demonstrated that this was due to enhanced antigen presentation by DCs [129]. In a model of HER2+ breast cancer, the combination of chemotherapy and a whole cell vaccine generated from tumor cells expressing the oncogene Neu and GM-CSF, delayed tumor growth by activating an anti-tumor T cell response [130]. These preclinical studies led to clinical testing of GM-CSF therapies with HER2 vaccines (summarized in Table 3). This combination has also been tested with the addition of trastuzumab, a HER2 targeting antibody [131,132,133]. Finally, GM-CSF delivery has also been achieved clinically using allogenic breast cancer cells engineered to secrete high levels of GM-CSF, termed SV-BR-1-GM cells [134,135] (NCT00095862). Another approach was to generate tumor cell vaccines from GM-CSF-expressing HER2+ breast cancer cells [136,137] (NCT00093834, NCT00399529). The results of these trials overall demonstrate improved response in TNBC patients with GM-CSF therapy, although the optimal combination with chemotherapy or other immunotherapies requires further preclinical and clinical testing. 

### 5.2. Macrophage Colony-Stimulating Factor (M-CSF)/Colony Stimulating Factor-1 (CSF-1)

#### 5.2.1. M-CSF and Breast Cancer Progression

M-CSF, also known as CSF-1, is an important mediator of monocyte recruitment, function and differentiation into macrophages. Two groups identified M-CSF as a secreted factor that was enhanced in breast cancer cells by EMT [23,37]. M-CSF was also increased with EMT in inflammatory breast cancer models [148]. Clinically, M-CSF and its receptor, Colony Stimulating Factor-1 Receptor (CSF1R), were elevated in breast cancer patients with local invasion or metastasis compared to those without tumor spread, and their expression predicted poor survival [149,150]. Further, serum levels of M-CSF were elevated in breast cancer patients with lymph node involvement compared to those without local invasion [151]. These findings suggest that M-CSF may support breast cancer metastasis and prompted one group to generate a M-CSF response signature composed of 603 genes. Applying this signature to breast cancer specimens, including those from early pre-malignant lesions know as ductal carcinoma in situ (DCIS), demonstrated that M-CSF activation was present in a subset of DCIS specimens [152,153]. Further, this signature was positively associated with tumor grade in malignant tumors. M-CSF activation may be present throughout several steps of carcinogenesis from malignant transformation through metastasis.

#### 5.2.2. Preclinical and Clinical Targeting of M-CSF in Breast Cancer

In several elegant in vivo studies undertaken in TNBC-like mouse models, M-CSF dynamically impacted the TME, largely through recruitment and education of TAMs. MMTV-PyMT mice containing null recessive CSF1 (encodes M-CSF) had no change in primary tumor incidence or growth but had decreased formation of lung metastases [154]. Follow-up experiments demonstrated that this is due to the impact of M-CSF on macrophage recruitment, which is required for the pro-metastatic angiogenic switch [155]. Intravital imaging revealed that macrophages utilize M-CSF to communicate with tumor cells via a paracrine loop of M-CSF secreted by tumor cells and EGF secreted by macrophages [156,157]. This signaling loop supported tumor cell migration and intravasation into the blood stream and thus metastasis. In another study, CSF1R was inhibited with a blocking antibody in pre-malignant MMTV-HER2 tumors that model HER2+ breast cancer [158]. This decreased lung metastasis but did not impact primary tumor growth. These promising preclinical experiments led to clinical targeting of either M-CSF or CSF1R in patients with advanced breast cancer (summarized in Table 3). However, results from these trials showed little clinical benefit. A lack of clinical response could be explained by a preclinical study where TNBC-like 4T1 mammary tumors were treated with several anti-M-CSF/CSF1R therapies [159]. Each treatment resulted in increased neutrophil and monocyte recruitment that together supported tumor progression. This suggests that the impact of M-CSF or CSF1R inhibition on cells other than macrophages should be evaluated. 

## 6. Other Secreted Immune Suppressive Factors Identified via EMT Modulation

In addition to altering tumor cell metabolism and cytokine secretion, our group and the Weinberg group identified other secreted factors, such as CHI3L1 and OPN, that are modulated by EMT in breast cancer models [23,30]. This section will discuss these factors as emerging immune modulatory targets in mTNBC even though strategies to target them are at an early stage of development or do not yet exist.

### 6.1. Chitinase-3 Like-1 (CHI3L1)

#### 6.1.1. CHI3L1 and Breast Cancer Metastasis

CHI3L1 (also referred to as YKL-40, human cartilage glycoprotein-39/HC gp-39 or murine breast regression protein-39/BRP39) is a chitinase-like protein that lacks enzymatic activity [160]. It is secreted by many cell types including activated macrophages [161,162], neutrophils [163], chondrocytes [164], osteoblasts [165], activated T cells [166] and cancer cells [167]. Serum CHI3L1 is upregulated in patients with chronic inflammatory diseases like asthma and chronic obstructive pulmonary disease (COPD) [168], and it is a biomarker of poor survival in various cancers (as reviewed in [169]). In the normal breast, CHI3L1 was increased during the dynamic remodeling that accompanies weaning known as involution [170]. In primary breast cancers, elevated levels of tumor and serum CHI3L1 were correlated with poor differentiation, mesenchymal markers, tumor grade and a shorter relapse-free survival [171,172,173]. In metastatic breast cancer patients, high serum CHI3L1 levels predicted a decrease in overall survival when compared to patients with normal CHI3L1 levels [174]. Thus, CHI3L1 may support metastatic breast cancer progression, and exploration of this idea has just begun in preclinical models. We demonstrated that human ER+ breast cancer cells with a point mutation in the ligand binding domain of *ESR1* (encodes ER) had increased expression of CHI3L1 when compared to those with wild type *ESR1* [175]. Increased CHI3L1 in mutant *ESR1* cells conferred an invasive advantage that was lost when cells were treated with a CHI3L1 blocking antibody. CHI3L1 also enhanced migration of the mesenchymal-like HER2 expressing breast epithelial cell line, D492HER2 [176]. In this study, CHI3L1 simultaneously acted in a paracrine manner to support angiogenesis. Further, blockade of astrocyte secreted CHI3L1 increased the survival of mice harboring cortical breast cancer metastases [177]. Together these studies suggest that CHI3L1 mediates a signaling loop between tumor cells and cells in the TME. However, the impact of EMT on this signaling loop remains underexplored. Preliminary evidence from prostate cancer and non-small cell lung cancer showed that CHI3L1 induces EMT markers such as TWIST, SLUG and SNAIL [178,179]. Thus, the role of CHI3L1 as a mediator of EMT needs to be investigated in breast cancer.

#### 6.1.2. Impact of CHI3L1 on the Tumor Microenvironment

In addition to its effects on breast cancer cells, CHI3L1 impacts immune cells. Treatment of macrophages with CHI3L1 increased pro-tumor M2 polarization genes and enhanced the infiltration of immune suppressive T cells into primary mammary tumors [180,181]. Alternatively, inhibition of CHI3L1 in mouse models of TNBC supported an active TME and decreased spontaneous and late-stage mammary carcinoma lung metastasis [181,182,183]. These studies call for further exploration of the impact of CHI3L1 on immune cells. For instance, single cell immunophenotyping technologies could be used to determine which cells in the TME produce the most CHI3L1 and which cells are the most impacted by CHI3L1. Such a study may be imperative before CHI3L1-directed therapies, such as CHI3L1 blocking antibodies, enter clinical testing in mTNBC.

### 6.2. Secreted Phosphoprotein 1 (SPP1)/Osteopontin (OPN)

#### 6.2.1. OPN and Breast Cancer Metastasis

SPP1 encodes the secreted integrin-binding glycol-phosphoprotein OPN. Circulating and tumor levels of SPP1 positively correlated with a poor prognosis and shortened survival in breast cancer patients [184,185]. Further, low OPN levels predicted a better response to standard of care breast cancer therapies such as neoadjuvant chemotherapy [186] and endocrine inhibitors [187]. Due to the predictive value of SPP1/OPN levels, new immunoassays are being developed to quantify OPN in breast tissues [188]. However, the predictive value of the OPN transcript and protein do not always correspond. For instance, only SPP1 gene expression, not protein expression, predicted recurrence following tamoxifen treatment [187]. Future studies should evaluate both mRNA and protein levels of this gene to fully understand the predictive value of OPN. Despite this potential limitation, SPP1 was highly expressed in mesenchymal-like aggressive breast cancers, such as those representing the basal-like subtype, when compared to more epithelial less-aggressive subtypes of the disease [189,190,191,192]. Interestingly, knockdown of SPP1 by the Weinberg group in MMTV-PyMT-derived cell lines significantly decreased mesenchymal markers such as Snail and Zeb1 [23], and this decreased metastatic potential [193]. Together, these studies suggest that OPN may function in an autocrine manner to support breast tumor cell EMT and metastasis, although further preclinical testing of this idea is needed.

#### 6.2.2. OPN and the Breast Cancer Microenvironment

Tumor-derived OPN also supports tumor progression in a paracrine manner. For instance, SPP1 knockdown in TNBC-like mammary carcinoma cells inhibited tumor growth in both immunocompetent and immunodeficient mice [23]. Interestingly, when tumor growth was compared between these experimental groups, SPP1 knockdown tumors grew better in immunodeficient compared to immunocompetent mice. These data suggest that the paracrine, immune modulatory functions of SPP1 contributes to breast tumor progression. For example, knockdown of SPP1/Opn in tumor cells promoted polarization of macrophages to an anti-tumor phenotype [23]. In addition to macrophages, T cells were impacted in this study which included increased T cell infiltration and cytotoxicity but decreased T cell suppression. Within the stroma, SPP1 induced fibroblast reprogramming and activation toward a pro-inflammatory, pro-tumor phenotype [194]. OPN even functions as an adhesive substrate for platelet aggregation [195], which has previously been shown to promote tumor progression and metastasis [196]. Together these studies suggest that tumor secreted-OPN may suppress the TME. 

OPN is also expressed by several cell types in the TME such as myeloid cells [193], fibroblasts [194] and in the case of bone metastases osteoblasts/osteoclasts [197]. Expression of OPN by these cells supports metastasis. This was demonstrated by a study that introduced TNBC-like 4T1 mammary tumor cells into SPP1^−/−^ null mice which resulted in decreased spontaneous lung metastasis [193]. Interestingly, simultaneous knockdown of SPP1 in 4T1 tumor cells caused a complete abrogation of metastasis. Therefore, SPP1 expressed in both the microenvironment and in the tumor may support breast cancer metastasis. Given these data, SPP1/OPN appears to be a promising therapeutic target in breast cancer. Unfortunately, no targeted therapies have been developed, which is likely due to the complexity of the varied functions of SPP1 throughout the body [198,199,200,201,202]. However, given the impact of SPP1/OPN on breast cancer progression in preclinical studies, the search for effective and safe SPP1/OPN targeted therapies should continue.

## 7. Conclusions and Future Directions

This review highlights how reversal of EMT in TNBC models revealed immune modulatory factors that are in preclinical or clinical development for treatment of mTNBC. Some of these targets, such as M-CSF, have been tested in multiple clinical trials and seen limited anti-tumor activity. While these results may seem discouraging, they suggest that additional studies are needed to better understand the complex microenvironment of metastatic breast cancers. For instance, preliminary analyses suggest that TNBC patients with chemotherapy-responsive, early-stage disease have an influx of anti-tumor immune cells after one course of chemotherapy [203]. Nonetheless, immune cell presence drops below baseline by the end of the chemo regimen. Whether metastases have a similar peak of immune response to chemotherapy remains unknown. Preclinical or clinical studies testing this idea may reveal a treatment window where immunotherapies are most effective. 

The heterogeneity of the immune cell milieu between metastatic sites also needs to be considered when developing additional TNBC immunotherapies. Preliminary studies suggest that liver metastases are more likely to be immune deserts than other metastatic sites [204]. Thus, patients with liver metastases may benefit from strategies that enhance immune cell presence, such as adoptive T cell transfer, rather than those that suppress immune cell recruitment like M-CSF blocking antibodies. As combination immunotherapy clinical trials become prevalent in breast cancer, more extensive immune profiling of metastatic sites may also reveal optimal immunotherapy combinations.

Finally, this review largely focuses on factors increased during EMT to promote a suppressed TME. Other breast cancer subtypes, such as ER+ disease, that undergo less of an oncogenic EMT progression, may not be responsive to the targets discussed in this review. Future studies should focus on immune modulatory pathways/proteins upregulated in epithelial compared to mesenchymal breast cancer models. These could serve as immunotherapy targets in other breast cancer subtypes that are less responsive to immunotherapies. Manipulation of EMT has and will continue to be a powerful tool to discover clinically relevant breast cancer immunotherapy targets. 

## Figures and Tables

**Figure 1 pharmaceuticals-14-01122-f001:**
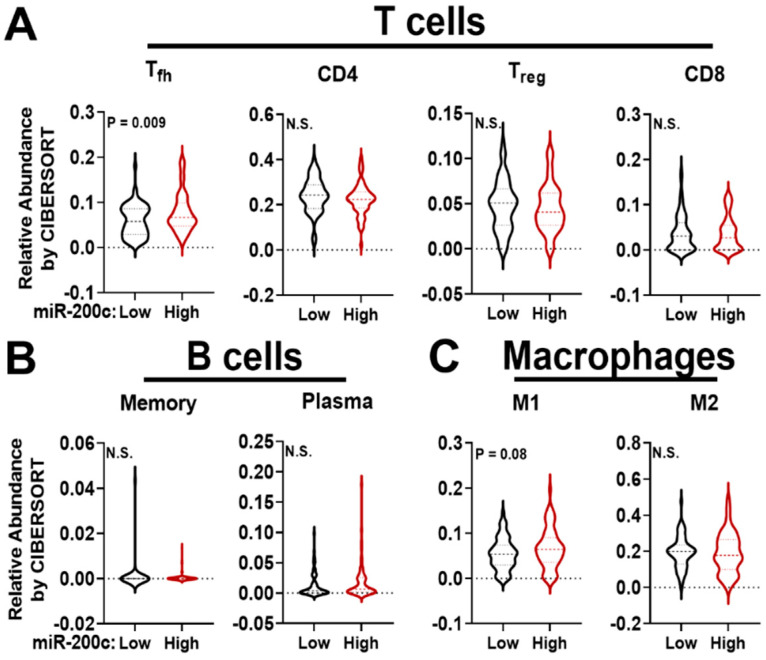
Relative immune cell abundance as predicted by gene expression using CIBERSORT on breast cancer specimens with high versus low miR-200c. mRNA profiling curated by The Cancer Genome Atlas (TCGA), Nature 2012 dataset [31] was assessed for the relative amounts of (**A**) T cells, (**B**) B cells, and (**C**) macrophages using CIBERSORT [32]. Specimens were stratified by expression of miR-200c in the lowest quartile (Low, N = 65) or highest quartile (High, N = 64). Shown is the median number of predicted immune cells. Student’s unpaired two-tailed t-test.

**Figure 2 pharmaceuticals-14-01122-f002:**
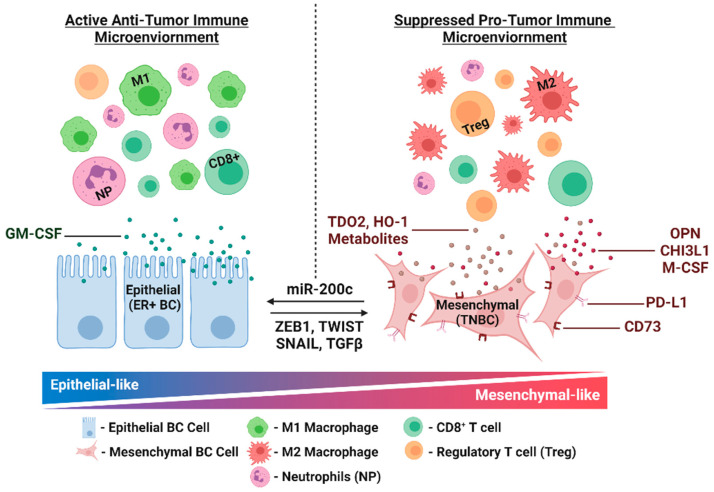
Model of actionable immune modulatory pathways revealed by epithelial-to-mesenchymal transition (EMT) manipulation in triple negative breast cancer (TNBC). EMT was manipulated in TNBC models by restoring miR-200c to mesenchymal-like TNBC cells or selecting for epithelial versus mesenchymal populations using expression of mesenchymal transcription factors (ZEB1, TWIST, SNAIL). These approaches led to identification of several immune modulatory factors (cytokines: GM-CSF, M-CSF; metabolizing enzymes: CD73, TDO2, HO-1; other secreted factors: CHI3L1, OPN; checkpoint protein: PD-L1). These factors may work in tandem to support a suppressed pro-tumor microenvironment in mesenchymal-like TNBC, and a more active anti-tumor microenvironment in epithelial-like ER+ breast cancers. This figure was made with biorender.com.

**Figure 3 pharmaceuticals-14-01122-f003:**
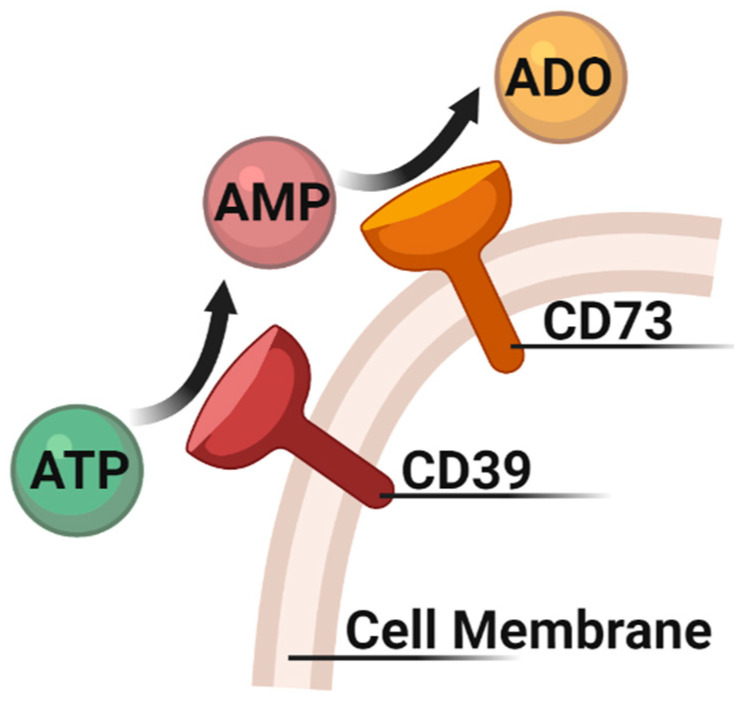
Adenosine metabolism pathway. Adenosine triphosphate (ATP) is degraded to adenosine monophosphate (AMP) by ectonucleoside triphosphate diphosphohydrolase-1 (CD39). AMP is then converted to adenosine (ADO) by 5’-Nucleotidase (NT5E/CD73). Both enzymes are expressed on the cell surface of numerous cells in the tumor microenvironment. This figure was made with biorender.com.

**Figure 4 pharmaceuticals-14-01122-f004:**
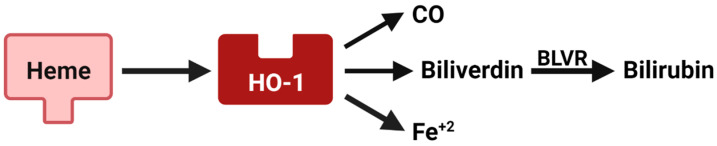
Heme metabolism pathway. Heme oxygenase-1 (HO-1) degrades heme into three immune modulatory byproducts, carbon monoxide (CO), ferrous iron (Fe^+2^) and biliverdin. Biliverdin is quickly converted to bilirubin via biliverdin reductase (BLVR). This figure was made with biorender.com.

**Table 1 pharmaceuticals-14-01122-t001:** Pathways negatively correlated with miR-200c in breast cancer specimens. Gene Set Enrichment Analysis (GSEA) was conducted on genes that correlate negatively with miR-200c in breast cancer specimens from The Cancer Genome Atlas (TCGA), Nature 2012 dataset [31]. The top GSEA Hallmark pathways are shown. Those involved in immune modulation are ***italicized and bolded***. ES = Enrichment Score; NES = Normalized Enrichment Score; NOM *p*-val. = Nominal *p*-value; FDR q-val. = False Discovery Rate q-value.

Top Pathways Downregulated by miR-200c					
Term	Count	ES	NES	NOM *p*-Val.	FDR q-Val.
HALLMARK_EPITHELIAL_MESENCHYMAL_TRANSITION	198	−0.75	−3.39	<0.00001	<0.00001
HALLMARK_MYOGENESIS	198	−0.54	−2.47	<0.00001	<0.00001
HALLMARK_KRAS_SIGNALING_UP	196	−0.54	−2.45	<0.00001	<0.00001
HALLMARK_UV_RESPONSE_DN	142	−0.56	−2.44	<0.00001	<0.00001
HALLMARK_COAGULATION	134	−0.56	−2.43	<0.00001	<0.00001
** *HALLMARK_TNFA_SIGNALING_VIA_NFKB* **	** *198* **	**−*0.53***	**−*2.40***	** *<0.00001* **	** *<0.00001* **
HALLMARK_ANGIOGENESIS	36	−0.68	−2.39	<0.00001	<0.00001
HALLMARK_APICAL_JUNCTION	195	−0.51	−2.31	<0.00001	<0.00001
** *HALLMARK_TGF_BETA_SIGNALING* **	** *54* **	** *−0.59* **	** *−2.21* **	** *<0.00001* **	** *<0.00001* **
** *HALLMARK_ALLOGRAFT_REJECTION* **	** *193* **	** *−0.48* **	** *−2.20* **	** *<0.00001* **	** *<0.00001* **
** *HALLMARK_INFLAMMATORY_RESPONSE* **	** *198* **	** *−0.48* **	** *−2.20* **	** *<0.00001* **	** *<0.00001* **
** *HALLMARK_COMPLEMENT* **	** *197* **	** *−0.47* **	** *−2.16* **	** *<0.00001* **	** *<0.00001* **
HALLMARK_APOPTOSIS	160	−0.48	−2.14	<0.00001	<0.00001
** *HALLMARK_IL2_STAT5_SIGNALING* **	** *195* **	** *−0.42* **	** *−1.93* **	** *<0.00001* **	** *0.000068* **
** *HALLMARK_INTERFERON_GAMMA_RESPONSE* **	** *192* **	** *−0.42* **	** *−1.93* **	** *<0.00001* **	** *0.000063* **
HALLMARK_HYPOXIA	199	−0.41	−1.90	<0.00001	0.000059
** *HALLMARK_IL6_JAK_STAT3_SIGNALING* **	** *83* **	** *−0.43* **	** *−1.76* **	** *<0.00001* **	** *0.000908* **
HALLMARK_ANDROGEN_RESPONSE	96	−0.41	−1.68	0.001431	0.002189
HALLMARK_P53_PATHWAY	194	−0.37	−1.67	<0.00001	0.002450
HALLMARK_BILE_ACID_METABOLISM	112	−0.39	−1.63	<0.00001	0.003382

**Table 2 pharmaceuticals-14-01122-t002:** Summary of clinical trials targeting the adenosine pathway in breast cancer.

Target	Drug	Clinical Trial Number	Study Phase	Cancer Type	Combination Therapy	Results; Publications
CD73	Oleclumab (MEDI9447)	NCT02503774	I	Solid tumors	PD-1	Ongoing; [64,65]
		NCT03616886	I/II	Inoperable or mTNBC	Paclitaxel, Carboplatin, PD-1	Recruiting
		NCT03875573	II	ER+ breast cancer	Doxorubicin-cyclophosphamide, pre-operative radiation	Ongoing; [66]
	LY3475070	NCT04148937	I	Advanced solid tumors	PD-1	Ongoing
CD73/A2AR	NZV930	NCT03549000	I/Ib	Advanced cancers	PD-1, A2AR	Recruiting
	CPI-006	NCT03454451	I/Ib	Select advanced cancers	PD-1 or A2AR	Recruiting; well-tolerated, some anti-tumor activity [67]
A2AR	NIR178	NCT03207867	II	Solid tumors	PD-1	Recruiting
A2AR+A2BR	Etrumadenant (AB928)	NCT03719326	I	mTNBC or ovarian	Doxorubicin and PI3Kγ or paclitaxel	Ongoing; favorable safety profile [68]
		NCT03629756	I	Advanced cancers	PD-1	

**Table 3 pharmaceuticals-14-01122-t003:** Summary of clinical trials targeting Granulocyte-Macrophage Colony-Stimulatory Factor (GM-CSF), Colony Stimulating Factor-1 Receptor (CSF1R) and Macrophage Colony-Stimulatory Factor (M-CSF) in breast cancer. NeuVax = HER2 vaccine with GM-CSF; DCIS = ductal carcinoma in situ; DFS = disease free survival; TAMs = tumor associated macrophages; PFS = progression free survival.

Target	Drug	Clinical Trial Number	Study Phase	Cancer Type	Combination Therapy	Results; Publications
GM-CSF	NeuVax vaccine	NCT01479244	III	HER2-low breast	-	Well-tolerated, no change DFS [138]
		NCT01570036	II	HER2-low breast	Trastuzumab	Well-tolerated, increased DFS in TNBC patients [139]
	Sargramostim	NCT00027807	I	Stage IV breast	IL-2 and autologous T cells	Increased cytotoxic T cells [140]
		NCT00524277	II	Breast	HER2 vaccine	Increased DFS for TNBC [141]
		NCT00436254	I	HER2+ breast and ovarian	HER2 vaccine	Immunogenic out to 60 weeks [142]
		NCT02636582	II	DCIS	HER2 vaccine	Ongoing
CSF1R	Pexidartinib (PLX-3397)	NCT01525602	I	Advanced solid tumors	Paclitaxel	Well-tolerated, promising decrease in monocyte recruitment [143]
		NCT01596751	I/II	Metastatic breast	Eribulin	Completed, results not published
		NCT01042379	II	Breast	-	Ongoing
	Emactuzumab (RG7155)	NCT01494688	I	Advanced or metastatic tumors	Paclitaxel	Decreased immune suppressive TAMs, no anti-tumor activity [144]
	LY3022855	NCT01346358	I	Advanced solid tumors	-	Dose dependent pharmacokinetics, no clinical activity [145]
M-CSF	Lacnotuzumab (MCS110)	NCT02435680	II	Advanced TNBC	Carboplatin and gemcitabine	On-target, no change in PFS [146]
		NCT02807844	Ib/II	Advanced tumors	PD-1	Anti-tumor response in pancreatic tumors [147]

## Data Availability

Data analyzed in this review can be access at: cbioportal.org.

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
