# Peer review of "Reversing an Oncogenic Epithelial-to-Mesenchymal Transition Program in Breast Cancer Reveals Actionable Immune Suppressive Pathways"

_pharmaceuticals, 2021, doi:10.3390/ph14111122_

Round 1

Reviewer 1 Report

This is a very good review article I had great pleasure to read. I liked that you focused on the multiple aspects that characterize EMT, including metabolism, without putting the different tumor cells into sub-groups like « claudin-low » or others… instead, you highlighted the plasticity of the tumor cells. You also gave clues on different ways to alter it, opening windows on future oncologic treatments. That’s refreshing!

Reviewer 2 Report

The Authors provide an extensive reviewe based on tackling the EMT response in TNBC. The work is very detailed and some paragraphs would require a bit of semplification/organization ot help the reader in following.  Minor revision required

Minor points: line 92 "immunocopormised" should be corrected
